# Fruit and Vegetable Concentrate Supplementation and Cardiovascular Health: A Systematic Review from a Public Health Perspective

**DOI:** 10.3390/jcm8111914

**Published:** 2019-11-08

**Authors:** Giulia Lorenzoni, Clara Minto, Maria Gabriella Vecchio, Slavica Zec, Irene Paolin, Manfred Lamprecht, Luisa Mestroni, Dario Gregori

**Affiliations:** 1Unit of Biostatistics, Epidemiology and Public Health, Department of Cardiac, Thoracic, Vascular Sciences and Public Health, University of Padova, 35131 Padova, Italy; giulia.lorenzoni@unipd.it (G.L.); slavica.zec@unipd.it (S.Z.); irene.paolin@studenti.unipd.it (I.P.); 2ZETA Research Ltd., 34129 Trieste, Italy; clara.minto@unipd.it (C.M.); MariaGVecchio@zetaresearch.com (M.G.V.); 3Otto Loewi Research Center, Division of Physiological Chemistry, Medical University of Graz, 8010 Graz, Austria; Manfred.Lamprecht@juiceplus.com; 4The Juice Plus+^®^ Science Institute, Collierville 38017, TN, USA; 5School of Medicine, University of Colorado, Aurora 80045, CO, USA; Luisa.Mestroni@ucdenver.edu

**Keywords:** fruits, vegetables, concentrates, supplementation, primary prevention

## Abstract

Fruits and vegetables (FV) are very important for the prevention of noncommunicable diseases (NCDs), but it has been demonstrated that FV consumption is below that recommended. Several companies have worked to offer FV concentrates, but it remains unclear whether they represent a potentially effective means of reducing the burden of NCDs. The present study provides a systematic review aimed at assessing the effect of FV concentrate supplementation on select parameters that are known to be risk factors for NCDs. The systematic review was done according to the PRISMA guidelines. Relevant studies were identified through the online databases PubMed, Scopus, Web of Science, and Embase. The physiological parameters of interest were total cholesterol, low-density lipoprotein, plasmatic homocysteine, systolic blood pressure, and body mass index. Data extraction was performed in duplicate. The results of the systematic review provided input for a Markov chain simulation model aimed at estimating the public health consequences of various scenarios of FV concentrate utilization on NCDs burden. The present results suggest a positive and significant role of FV concentrate supplementation on select parameters known to affect the risk of NCDs. Such an effect might be hypothesized to turn into mitigation of the burden of those NCDs modulated by the physiological parameters analyzed in the present systematic review.

## 1. Introduction

Over the past 20 years, the world has experienced a “nutrition transition” that is characterized by a shift in diet and physical activity patterns. The consumption of the ’Western diet’, which is rich in refined foods that are high in saturated fats and sugar and low in other nutrients, such as fibers and vitamins, alongside a reduction in time spent performing physical activity has led to a change in nutritional outcomes [1] leading to an epidemic of obesity, overweight, diabetes and other noncommunicable diseases (NCDs) [2]. NCDs include the “big four”—cardiovascular diseases (CVDs), diabetes, cancers, and chronic respiratory diseases—and others such as hepatic, gastroenterological, oral, Alzheimer’s, and Parkinson’s diseases. According to the WHO, NCDs were responsible for 60% and 68% of deaths worldwide in 2000 and in 2012, respectively [3]. The WHO reports that cardiovascular diseases account for 17.5 million people annually, followed by cancers (8.2 million), respiratory diseases (4 million), and diabetes (1.5 million) [4]. The causes of NCDs are multifactorial, varying from nonmodifiable (age, genetic predisposition, gender, and race) to modifiable factors (lifestyle). Initiatives that target modifiable risk factors, such as healthy lifestyle habits and a balanced diet, represent the first step in the prevention of NCDs. Specifically, the consumption of fresh fruits and vegetables is very important for the maintenance of a healthy lifestyle, and it has been widely demonstrated that their intake results in several beneficial effects [5,6], especially for what concerns cardiovascular health [7,8]. Fruits and vegetables provide a wide variety of different micronutrients and bioactive compounds such as vitamins, minerals, phytochemicals, and fibers, each of which has demonstrated a risk-reducing effect both in morbidity and mortality [9,10,11,12]. A recent WHO/FAO expert consultation report set the recommended intake of a minimum of 400 g fruits and vegetables per day for the prevention of chronic diseases [13]. Despite the recommendations, the consumption of fruits and vegetables continues to be below the standard recommendation [14,15], even though several strategies have been put forward in different countries to encourage people to eat these foods [16,17]. Several reasons, such as fruit and vegetable availability, taste and organizational difficulty due to work commitment, and unhealthy lifestyle habits [18,19], have been claimed to contribute to the poor achievement of the recommendations. Not least, to get fresh fruits and vegetables is a challenge in many parts of the world. Supplementation with vitamins and minerals would partly overcome these issues, especially in high-risk population subgroups [20], even though it is worth pointing out that the supplementation cannot replace the consumption of fresh fruit and vegetables completely.

Several companies have worked to offer fruit and vegetable concentrates that contain a well-balanced mixture of phytonutrients, vitamins, minerals, and other bioactive compounds from several fruits and vegetables to consumers. Although numerous studies have been conducted to estimate the effect of the concentrates on human metabolism [21,22,23,24], it remains unclear whether they represent a potentially effective means of reducing the burden of NCDs.

The objective of the present study was to perform a systematic review—according to the PRISMA guidelines [25]—with the aim of assessing the effects of fruit or vegetable concentrate supplementation on select physiological parameters that are considered relevant risk factors for the development of CVDs. The public health consequences of various scenarios of concentrate utilization are also examined in this paper.

## 2. Materials and Methods

A literature review was performed on the effect of fruit or vegetable concentrates on select physiological outcomes (total cholesterol (TC), low-density lipoprotein (LDL), plasmatic homocysteine (HCY), systolic blood pressure (SBP), body mass index (BMI), and TNF-α). The Preferred Reporting Items for Systematic reviews and Meta-Analyses (PRISMA) [25] was used for guiding the review process. The PRISMA flow-chart is provided in Figure 1.

Table 1 reports the PICOS criteria (participants, interventions, comparisons, outcomes, and study design) used to define the research question.

### 2.1. Information Sources

Relevant studies were identified through the online databases PubMed, Scopus, Web of Science, and Embase by collecting all available information published until 22 June 2018. We used the three keywords fruits, vegetables, and concentrates, as reported in Appendix A.

Each search string was adapted consistently with specific codification rules for each single database, and the results were limited to the human population. Moreover, we screened reference lists of eligible papers in order to identify further relevant studies.

### 2.2. Eligibility Criteria

Studies were considered eligible if they enrolled adult participants (≥18 years old) of both sexes with any of these clinical conditions: healthy, hypertension, or metabolic syndrome. Thus, all patients who were defined as healthy, healthy smokers, or healthy nonsmokers were included in the group “healthy”, while all patients who were affected by obesity and/or dyslipidemia were included in the category “metabolic syndrome”. Finally, we separately analyzed patients with hypertension to identify specific effects of fruits/vegetable concentrates on these types of subpopulations.

Relevant studies should test the supplementation with (i) fruits or (ii) vegetables or (iii) fruit and vegetable concentrates. According to the European council directive 2001/112/EC, a concentrate is considered a product that is obtained from fruit or vegetable juice by the physical removal of a specific proportion of the water content [26,27]. There was no limitation in terms of dosage or follow-up duration, whereas only oral intake was considered eligible for the present study.

In addition to that, we only included studies with at least one active treatment and one control condition (placebo treatment or no treatment). One-arm trials and studies without a control comparison were excluded.

Studies were considered eligible if they reported information on the effect of each concentrate on at least TC, LDL, HCY, SBP, BMI, and TNF-α. The full texts were analyzed by two independent reviewers, and articles with sufficient information about at least one outcome of interest were included. No additional information on unpublished data was requested from the authors of the papers.

No specific limitations on the study characteristics were applied: trials with randomized or nonrandomized, controlled, blinded, or nonblinded designs were included. Pilot studies, case reports, case series, one-arm trials, and papers in a language different than English were excluded. No restrictions were applied to the publication date or type of setting.

### 2.3. Study Selection

Two independent investigators worked in pairs and independently assessed the eligibility of articles following the step-by-step process of the PRISMA flow diagram [25]. If both reviewers considered the study eligible, the full text was obtained, and the investigators extracted the numerical and descriptive information. Any disagreement was discussed and solved through consensus.

### 2.4. Data Collection Process

For each eligible study, we collected information on the publication date, study design, type of population (healthy or unhealthy), age, smoking habits, treatment groups, follow-up duration, and changes in clinical parameters. For each outcome, we extracted data on baseline and follow-up values and/or data on the mean change from baseline. Data can be expressed as numerical and/or graphical information; in this latter case, we extracted results from graphs by using the procedure (adapted) from Guyot et al. [28]. Data extraction was performed in duplicate.

### 2.5. Estimation of Public Health Effects

To assess the public health consequences of various scenarios of concentrate utilization, we simulated the effectiveness of concentrate supplementation by linking the direct effect of each concentrate on a clinical surrogate (TC, LDL, HCY, SBP, BMI, and TNF-α) with the effect of the same surrogate on Coronary Heart Disease (CHD), Heart Failure (HF), or stroke. This was done because we found scarce empirical evidence about the direct effect of concentrates on primary outcomes, such as CVD morbidity.

For the linkage between each concentrate and the clinical surrogates of interest, data from nutritional trials were used. If more than one paper was available for the same population, the same outcome, and the same product, then only the study that reported the most effective action of the concentrate was considered. Similarly, if more than one active arm was included in the same study for the same type of population, the most effective arm was analyzed. For linkage between clinical surrogates and CHD/HF/stroke, data from a recent meta-analysis reporting the effect of TC, LDL, HCY, SBP, BMI, or TNF-α on at least one of the three main outcomes (CHD, HF or stroke) were used. Due to the lack of usable data, we assumed correspondence between morbidity incidence (number of subjects affected by a specific disease) and events (number of events such as myocardial infarction, stroke, etc.). Moreover, when information on total events or incidence was not available, we used data on death risk for the specific disease.

### 2.6. Statistical Methods for the Estimation of Public Health Effects

Statistical analysis is based on Markov Chain Monte Carlo (MCMC), which is a widely used stochastic procedure that properly accounts for uncertainty by repeatedly generating random samples that characterize the distribution of parameters of interest [29,30]. Disease prevalence was modeled according to a Pert distribution [31] (mean values and standard deviations set according to data described in Section 2.7). The effects of the considered biomarkers were simulated according to a Multivariate Normal (MVN) distribution [32], allowing a correlation parameter of 0.3 among them, symmetrically for each biomarker pair. To account for uncertainty in setting the correlation parameter, it was included as a parameter in the simulated MVN as the results of a random draw from a Uniform distribution on the segment [0.2–0.4]. The final impact of modifications in surrogate biomarkers on the outcome of interests was modeled using a Pert distribution, with means and standard deviations described in Table 2. Ten-thousand runs over 1000 simulations were conducted and eventually averaged, using the Mersenne-Twister algorithm [33].

The model was eventually stratified by age group and replicated for vulnerable groups to simulate and forecast the morbidity (total prevalence) and cost (total 10 years direct cost) for the time frame of 2015 to 2025. Markov chain 95% credibility intervals were derived for each point estimate of total cost and absolute prevalence for each subgroup. We programmed our model in ModelRisk 2016 (http://www.vosesoftware.com/). We present the difference in the prevalence and total direct cost in terms between no action taken and concentrate supplementation using different concentrates found in the literature. Our basic assumption is that the concentrate yields an effect (suggested by the literature) that remains stable in the 10-year period time frame.

Next, we assessed the effect of the administration of select concentrate supplementation on the population (or its subgroups). On the basis of the available evidence, relative risk reduction has been used to simulate different scenarios of supplementation, both on the whole population and on specific age groups and vulnerable populations. This allows deeper insight into how to identify the main target of the supplementation, with the aim of maximizing the gain in terms of CHD, HF, or stroke prevalence and the corresponding cost reduction.

### 2.7. Scenarios Considered for Health Impact Assessment

Different scenarios are projected into the 10-year time frame. The first scenario is that without any supplementation of any population. The second scenario is that regarding concentrate supplementation. Evidence of the effect on the following concentrates was found in the scientific literature: encapsulated fruit and vegetable concentrate, artichoke leaf juice and Jerusalem artichoke juice, orange juice, fruit and vegetable juice, cherry juice, and generic fruit juice.

Projections of CHD, HF, or stroke prevalence and direct cost were built for the United States as follows. We used the population projections for 2025 from the US census data, which takes into account the aging structure and demographic change of the existing population [41] and we have estimated disease prevalence by age group (18–24, 25–44, 45–64, 65–84, and 85+). CHD, HF, or stroke prevalence estimates were generated using data from the National Health Interview Survey [42]. Following the strategy of Heidenreich et al. [43], we assume that disease prevalence remains stable in time.

We replicated the analysis for vulnerable groups, where the following vulnerable groups were identified: (i) population with metabolic syndrome and (ii) population with hypertension. Where available, we have estimated the cost and prevalence of each of the abovementioned morbidity outcomes.

Average per capita morbidity cost was simulated using reference CVD cost estimates from Heidenreich et al. [43]. These estimates are based on the Medical Expenditure Panel Survey, and the authors provide projections up to 2030. We have used their 2015 projections of total direct cost and total prevalence in order to generate per capita direct cost. Due to a lack of reliable data on age-specific cost, CHD, HF, and stroke average costs were assumed to be constant across the age groups and vulnerable groups.

## 3. Results

### 3.1. Study Selection and Characteristics

Initially, 2378 records were retrieved: 310 from PubMed, 1287 from Scopus, 232 from Web of Science, and 549 from Embase. Title/abstract screening and full text reading led to the inclusion of 13 articles (Figure 1) that involved eight different products (encapsulated fruit and vegetable concentrate, orange juice, fruit and vegetable juice, fruit juice, cherry juice, Jerusalem artichoke juice, and artichoke leaf juice). The main characteristics of the studies included in the analysis are reported in Table 3.

We used data extracted from papers to estimate the effect size of each product on a specific clinical surrogate, following the diagram reported in Appendix A. Furthermore, the composition of each concentrate under study has been reported in Appendix A.

To simulate the effect of concentrates on the main outcomes (CHD, HF, and stroke), we linked clinical surrogates to the disease using risk data of recent meta-analyses [34,35,36,37,38,39,40], as reported in Table 2.

### 3.2. Impact of FV Concentrate Supplementation on Public Health

The results in terms of avoided CHD, stroke, and HF events and the corresponding total costs projected to 2025 are presented in Table 4 and Figure 2 (general population), Table 5 and Figure 3 (population with hypertension), and Table 6 and Figure 4 (population with metabolic syndrome).

For what concerns the general population (Table 4), encapsulated fruit and vegetable concentrate supplementation was found to have the strongest impact compared to other types of supplementation, especially for the prevention of CHD cases and related cost savings. The supplementation with encapsulated fruit and vegetable resulted in the prevention of 62.41 (95% C.I. 55.7–81.12) million CHD events corresponding to a cost saving of 109.76 (95% C.I. 96.17–153.44) billion dollars through the mediation of HCY. Similar results were obtained in the subset of the population affected by hypertension (Table 5); supplementation with encapsulated fruit and vegetable concentrate showed a strong impact on the number of CHD events mediated by HCY, 46.49 (95% C.I. 40.6–64.47) million CHD events avoided with a cost saving of 137.27 (118.17–202.7) billion dollars). Conversely, supplementation with concentrates of the population affected by metabolic syndrome had only a small effect on the prevention of CHD/HF/stroke events (Table 6), as is clearly shown in Figure 4. In this case, orange juice concentrate seemed to be effective in preventing CHD cases, which resulted in the prevention of 29.12 (95% C.I. 18.34–57.4) and 22.47 (13.5–49.67) million cases through the effect on TC and LDL, respectively, in 2025.

## 4. Discussion

The objective of the present study was to perform a systematic review on the effect of fruit or vegetable concentrate supplementation on select physiological parameters that are considered to be relevant risk factors for the development of CVDs. In addition, the potential public health impact of long-term supplementation with such fruits and/or vegetable concentrates was simulated. Since we did not consider indirect cost, these results represent a lower bound of the expected total costs due to CHD, HF or stroke, and the corresponding potential savings due to fruit/vegetable concentrate supplementation.

Even though a conservative approach was adopted, our results suggest a positive and significant role of supplementation on the public health burden of CVDs. The supplementation with encapsulated fruit and vegetable concentrate was found to have the strongest effect on the reduction of CHD cases through HCY in both the general population and in the population affected by hypertension. Conversely, in the population with metabolic syndrome, orange juice was found to have the strongest impact on the reduction of the burden of CVDs, especially for what concerns CHD. The effect of the supplementations considered in the analyses was less pronounced for stroke and HF compared to CHD in all the populations considered (general, with hypertension, and with metabolic syndrome).

We estimated the effectiveness of supplementations by evaluating their action on specific biomarkers of metabolic and cardiac function. Such choice is motivated by the fact that there is growing evidence on the effect of nutrient intake on circulating biomarkers [53]. All the biomarkers considered in the analysis have been found to be involved in the onset of the chronic diseases of interest. TC, LDL, and hypertension have been widely recognized by the scientific literature as key factors contributing to the pathogenesis of cardiovascular disorders [54,55]. In addition, BMI is a strong predictor of chronic conditions affecting the metabolic and cardiovascular systems [56]. TNF-α has been shown to be involved in the pathogenesis of atherosclerosis since it promotes the recruitment of inflammatory cells and blood vessel remodeling [57,58]. In addition, HCY is an independent risk factor for atherosclerosis. It can alter arterial structure and function by increasing the proliferation of smooth muscle cells, oxidative stress, and endothelial dysfunction [59]. In a recent study on subjects who underwent angiographic examination, the level of serum HCY was higher in patients with coronary artery disease than in healthy subjects [60]. Such evidence supports the findings of the present study, showing that the strongest effect of the supplementation (with encapsulated fruit and vegetable concentrates) was on CHD cases, through the modulation of HCY.

However, it is worth pointing out that, due to the small number of scientific trials on concentrate products, the effect of these supplementations on biomarkers and primary cardiovascular disorders cannot be confirmed with certainty. For what concerns HCY that was found to be the channel through which the supplementation with encapsulated fruit and vegetable showed the most relevant effect on the burden of CHD, evidence in the field is controversial. In a cross-sectional study on elderly people, the consumption of fruits and vegetables was found to be associated with lower levels of total plasma HCY [61]. Conversely, a randomized clinical trial on healthy women found out that the increased consumption of fruit and vegetable did not affect HCY concentration [62]. Similar results have also been obtained in a recently published randomized controlled trial [63]. However, it is worth pointing out that it is difficult to compare results from such studies since different study designs have been employed; in addition to that, different levels of fruit and vegetable consumption have been tested in populations with different characteristics.

### Study Limitations

Two studies associated the fruit and/or vegetable concentrate to a specific diet: van den Berg includes one controlled meal with a vegetable burger, while patients enrolled by Antal had a low-fatty-diet prescription [22,51]. Consequently, we cannot rule out that the effect of such two products (fruit juice and Jerusalem artichoke concentrate) could be modulated by the additive action of controlled diets or by the content of other compounds than micronutrients. Regarding population characteristics, one study evaluated the effect of orange juice by comparing an active-treatment group composed of healthy subjects with a placebo group that included both healthy and unhealthy subjects affected by hypercholesterolemia [44]. In this case, the estimated effect of orange juice administration could be influenced by the heterogeneity of study samples. Not all studies included a double-blind design, and most of them provided fruit and/or vegetable supplementation for less than two months.

The inclusion of some products, noticeably the Jerusalem artichoke, in the review, might be questionable since a prebiotic effect might be claimed for them. The choice of including or not a concentrate in the review was based on the official definition of concentrate provided in the European council directive 2001/112/EC. According to such criteria, the Jerusalem artichoke was eligible to be included in the review. Not least, a clear definition of prebiotics is missing. Recently, the International Scientific Association for Probiotics and Prebiotics (ISAPP) published a consensus statement providing a revised definition of prebiotics [64]. However, the U.S. Food and Drug Administration (FDA) and the European Food Safety Authority (EFSA) do not have established an official definition for prebiotics yet. This is a matter of confusion for researchers, the food industry/market, and consumers [65]. In the present case, the lack of an official definition of prebiotics provided by official regulatory authorities does not allow for clearly distinguishing prebiotics from concentrates, thus justifying the exclusion of Jerusalem artichoke from the review.

Another study limitation is represented by the fact that different types of concentrates were included (i.e., both single and mixed concentrates), each one with a different mechanism of action on the biomarkers of interest, providing different amounts of phytonutrients, and both encapsulated and not encapsulated concentrates were considered. Such an issue was taken into account in the analyses, and only single estimates for each class of products were provided instead of pooled ones.

Finally, it is worth noting that studies considered in this review are based on very small sample sizes, reflecting a lack of current research on concentrates. No long term, large scale-controlled studies are available in this area and specifically for targeted populations.

## 5. Conclusions

Present results suggest that the supplementation of the population with encapsulated fruit and vegetable (general population and population with hypertension) and orange juice (population with metabolic syndrome) would result in a reduction of the burden of CVDs, especially of CHD cases and of related direct costs. Such promising results suggest the need and the opportunity to further investigate the properties of concentrate products to discover the potential health impacts of different groups of nutraceutical compounds.

## Figures and Tables

**Figure 1 jcm-08-01914-f001:**
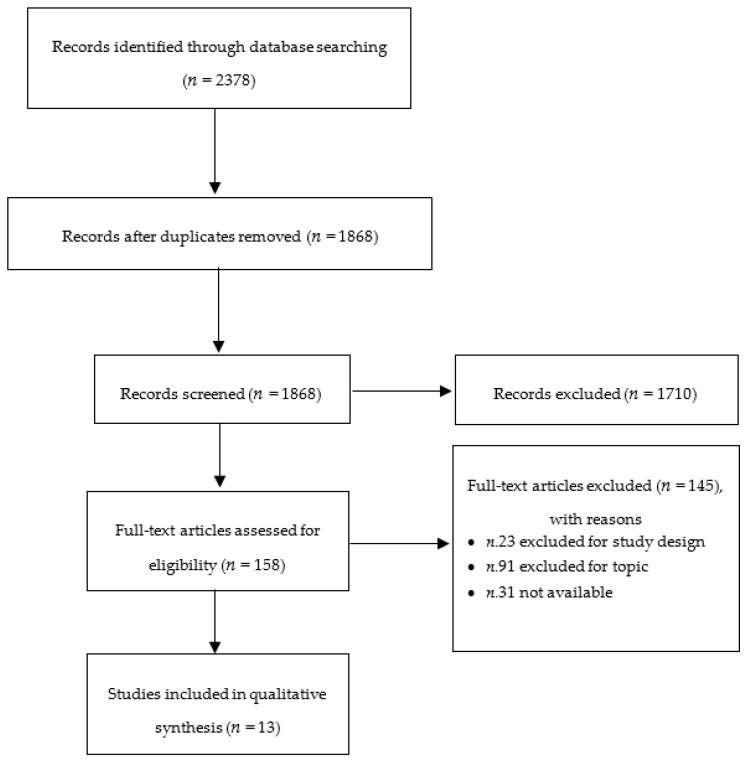
Flow chart of the literature selection.

**Figure 2 jcm-08-01914-f002:**
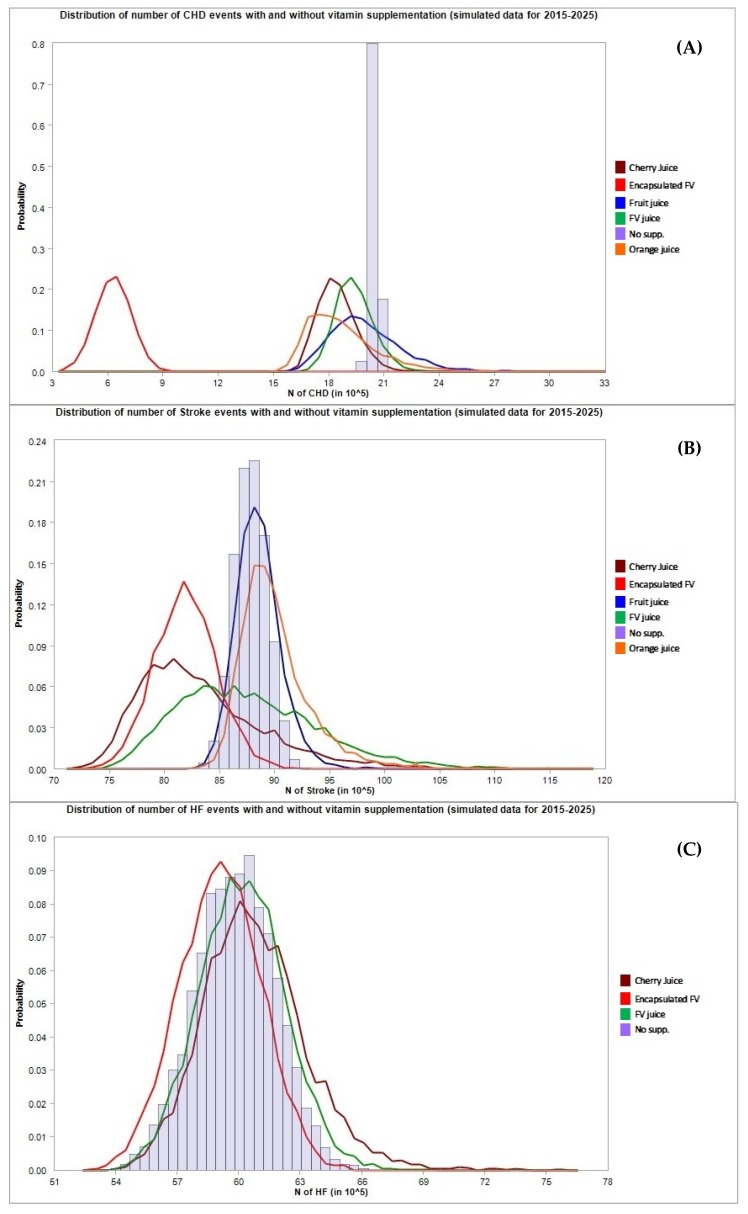
Number of cardiovascular events avoided under different supplementation regimes (estimates for the general population). The first chart (**A**) refers to CHD cases, the second chart (**B**) refers to stroke cases, the third chart (**C**) refers to HF cases.

**Figure 3 jcm-08-01914-f003:**
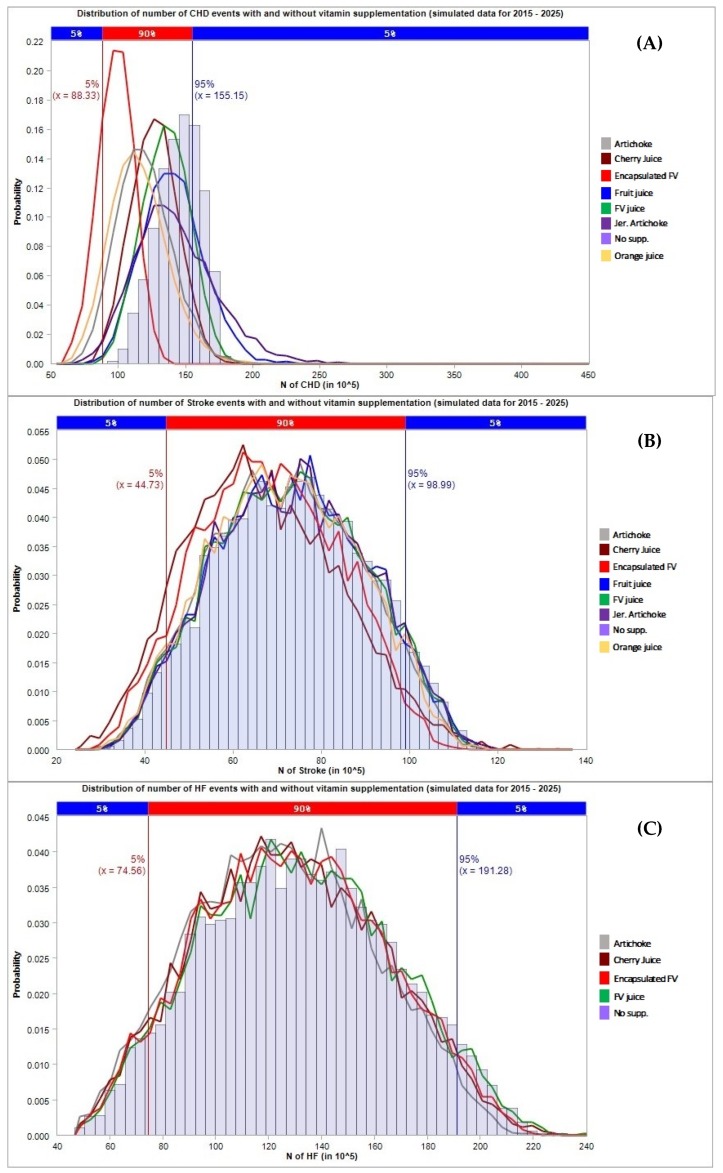
Number of cardiovascular events avoided under different supplementation regimes (estimates for the population with hypertension). The first chart (**A**) refers to CHD cases, the second chart (**B**) refers to stroke cases, the third chart (**C**) refers to HF cases.

**Figure 4 jcm-08-01914-f004:**
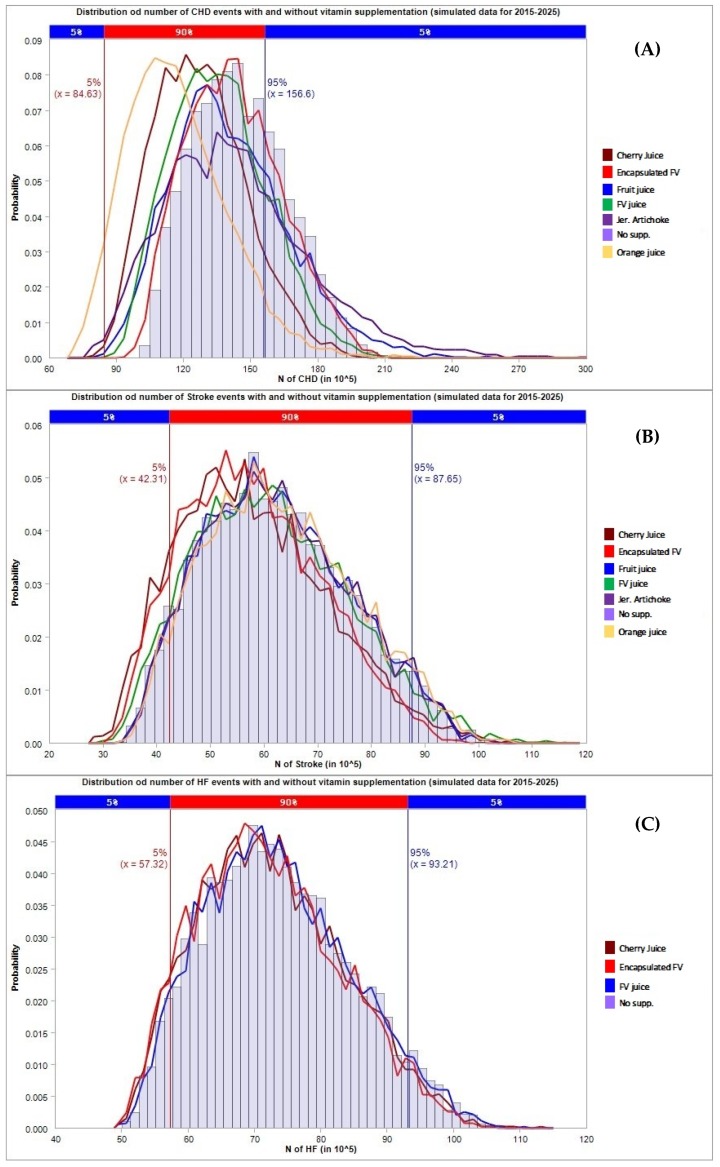
Number of cardiovascular events avoided under different supplementation regimes (estimates for the population with metabolic syndrome). The first chart (**A**) refers to CHD cases, the second chart (**B**) refers to stroke cases, the third chart (**C**) refers to HF cases.

**Table 1 jcm-08-01914-t001:** PICOS criteria (participants, interventions, comparisons, outcomes, and study design) used to define the research question.

Participants	Humans aged ≥ 18 years of both sexes with any of these clinical conditions: healthy, hypertension or metabolic syndrome
Interventions	Supplementation with (i) fruits or (ii) vegetables or (iii) fruit and vegetable concentrates
Comparisons	Placebo or No treatment
Outcomes	Total cholesterol; Low-density lipoprotein; Plasmatic homocysteine; Systolic blood pressure; Body mass index
Study Design	Interventional

**Table 2 jcm-08-01914-t002:** Articles used to estimate the effect size of a single concentrate product on a specific clinical surrogate.

Surrogate	Link to Main Outcome	References
TC	1 mmol/L lower TC is associated with lower CHD mortality equal to:	Prospective Studies Collaboration 2007 [34]
	- hazard ratio 0.44 (0.42–0.48) in both sexes at ages 40–49	
	- hazard ratio 0.66 (0.65–0.68) in both sexes at ages 50–69	
	- hazard ratio 0.83 (0.81–0.85) in both sexes at ages 70–89	
LDL	For a 10 mg/dL (0.26 mmol/L) reduction:	Briel et al. 2009 [35]
	- relative risk reduction is 7.1% (4.5% to 9.8%) for CHD events	
HCY	For each 5 μmol/L increment:	Peng et al. 2015 [36]
	- pooled risk ratio is 1.52 (1.26–1.84) for CHD deaths	
SBP	Every 10 mmHg reduction:	Ettehad et al. 2016 [37]
	- reduced the CHD events (relative risk (0.83, 0.78–0.88)	
	- reduced the STR events (relative risk (0.73, 0.68–0.77)	
	- reduced the HF (relative risk (0.72, 0.67–0.78)	
BMI	The relative risk for a 5 unit increment is:	Aune et al. 2016 [38]
	- 1.41 (1.34–1.47) for HF incidence	
TNF-α	The increase of 0.668 pg/mL in TNF-α is equal to an increase of STR risk with an odds ratio of 1.813 (1.194–2.748)	Dong et al. 2015 [39]
	1-SD increment of TNF-α is associated with increased risk of CHD:	
	- hazard ratio 1.09 (0.92–1.30)	Kaptoge et al. 2014 [40]

**Table 3 jcm-08-01914-t003:** Characteristics of the articles that were included in the analysis.

Author, Year	Study Design	Population Healthy/Unhealthy	Smoking Habits	Age (Mean ± SD)	Treatment Groups	Follow-Up	Clinical Parameters
Ali et al. 2011 [21] #	Randomized, double-blind, placebo-controlled, crossover study	Males and Females Metabolic syndrome	Non smokers	56.9 ± 11.2 56.9 ± 11.2 56.9 ± 11.2	Encapsulated FV concentrate (*n* = 60) Encapsulated FV concentrate and berry juice (*n* = 60) Placebo (*n* = 60)	8 weeks	TC, LDL, BMI
Antal et al. 2008 [22]	Parallel-group trial	Female Obese	-	53.0 ± 7.62 40.3 ± 8.89	Jerusalem artichoke (*n* = 17) Placebo (*n* = 6)	12 weeks	TC, LDL
Cesar et al. 2010 [44] *	Placebo-controlled study	Males and females Healthy or Hypercholesterolemia	-	35.8 ± 11.6 44.0 ± 12.0	Orange juice (*n* = 31) No treatment (*n* = 8)	60 days	TC, LDL, BMI
Cesar et al. 2010B [44] *	Placebo-controlled study	Males and females Hypercholesterolemia	-	42.3 ± 14.2 44.0 ± 12.0	Orange juice (*n* = 14) No treatment (*n* = 8)	60 days	TC, LDL, BMI
George et al. 2012 [45]	Randomized, single-blind, controlled, crossover dietary intervention	Males and females Healthy	-	45.0 ± 10.0 45.0 ± 10.0	Fruit and vegetable (*n* = 39) Placebo (*n* = 39)	6 weeks	TC, LDL, HCY, SBP, BMI
Lamprecht et al. 2007 [23]	Randomized, double-blind, placebo-controlled study	Male Healthy	-	34.3 ± 5.1 33.8 ± 5.7	Encapsulated FV concentrate (*n* = 21) Placebo (*n* = 20)	28 weeks	TNF-α
Lamprecht et al. 2013 [24]	Randomized, double-blind, placebo-controlled study	Females Obese	Non smokers	40.8 ± 3.7 41.3 ± 4.2	Encapsulated FV concentrate (*n* = 21) Placebo (*n* = 21)	8 weeks	TNF-α
Lynn et al. 2014 [46]	Parallel, open-label study	Males and females Healthy	Non smokers	38.3 ± 6.16 37.2 ± 5.78	Cherry juice (*n* = 24) Placebo (*n* = 19)	6 weeks	TC, SBP
Novembrino et al. 2011 [47]	Randomized, double-blind, placebo-controlled study	Males and females Healthy	Smokers	46.6 ± 7.9 51.4 ± 12.0	Encapsulated FV concentrate (*n* = 26) Placebo (*n* = 25)	3 months	TC
Panunzio et al. 2003 [48]	Randomized, crossover study	Males and females Healthy	-	20–56 20–56	Encapsulated FV concentrate (*n* = 14) Placebo (*n* = 10)	4 weeks	HCY
Plotnick et al. 2003 [49]	Randomized, double-blind study	Males and females Healthy	Non smokers	-	Encapsulated FV concentrate (*n* = 14) Placebo (*n* = 10)	4 weeks	LDL
Roghani-Dehkord et al. 2009 [50] §	Randomized, double-blind, placebo-controlled study	Males Middle hypertension	Non smokers	43.8 ± 8.28 44.1 ± 8.74 43.7 ± 7.47	Artichoke Leaf Juice 100 mg (*n* = 35) Artichoke Leaf Juice 50 mg (*n* = 39) Placebo (*n* = 33)	12 weeks	TC, LDL, SBP, BMI
Van Den Berg et al. 2001 [51]	Randomized, open label, placebo controlled, crossover trial	Males Healthy	Smokers	33.0 ± 11.0 33.0 ± 11.0	Fruit juice (*n* = 22) Placebo (*n* = 22)	3 weeks	TC, SBP
Williams et al. 2017 [52]	Randomized, double-blind, placebo-controlled study	Males and females Overweight/Obese	Non smokers	61.4 ± 1.5 57.9 ± 1.4	Encapsulated FV concentrate (*n* = 28) Placebo (*n* = 28)	8 weeks	TC, LDL, SBP, BMI, TNF-α

* In his original study, Cesar included three treatment groups: one placebo group, one active group with healthy subjects and one active group with dyslipidemic subjects. For this reason, we repeated the articles in two different rows. # In his original study, Ali included three treatment groups: one placebo group, one group of subjects who were treated with encapsulated fruit and vegetable concentrate, and one group of subjects who were treated with encapsulated fruit and vegetable concentrate and berry juice. In our analysis, all active arms were considered equal. The data on encapsulated fruit and vegetable concentrate were used to estimate the effect size of total cholesterol (TC), while the data on encapsulated fruit and vegetable concentrate and berry juice were used to estimate the effect sizes of low-density lipoprotein (LDL) and BMI. § In his original study, Roghani-Dehkord included three treatment groups: one placebo group, one group of subjects who were treated with 100 mg artichoke juice, and one group of subjects who were treated with 50 mg artichoke juice. In our analysis, all active arms were considered equal and named artichoke leaf juice. The data for 100 mg artichoke juice were used to estimate the effect size of systolic blood pressure (SBP), while the data for 50 mg artichoke juice were used to estimate the effect size of TC, LDL, and BMI.

**Table 4 jcm-08-01914-t004:** Estimated effect in terms of the absolute numbers of event reductions and their associated direct costs for different supplementation regimes in the general population, projected in 2025. The median and the 95% credibility interval are reported. Reported events are expressed in millions of events (95% C.I. lower bound; median; 95% C.I. upper bound), while direct costs are expressed in billions of dollars (95% C.I. lower bound; median; 95% C.I. upper bound).

		Coronary Heart Disease		Stroke		Heart Failure	
		Events	Direct Costs	Events	Direct Costs	Events	Direct Costs
Encapsulated fruit and vegetable	TC	(29.97, 36.26, 53.72)	(52.16, 63.65, 100.01)	(1.26, 1.98, 4.76)	(4.42, 6.91, 17.26)		
	LDL	(0.17, 0.61, 1.9)	(0.29, 1.07, 3.44)				
	HCY	(55.7, 62.41, 81.12)	(96.17, 109.76, 153.44)				
	SBP	(6.47, 8.87, 16.17)	(11.34, 15.54, 29.6)	(4.38, 6.02, 10.66)	(15.13, 21.07, 39.03)	(0.45, 0.71, 1.82)	(1.56, 2.49, 6.7)
	TNF-α	(1.17, 1.85, 3.67)	(2.05, 3.23, 6.7)	(10.34, 11.21, 13.5)	(35.21, 39.14, 52.25)		
Orange juice	TC	(7.04, 21.08, 42.86)	(12.37, 37.11, 78.92)	(0.26, 1.24, 5.8)			
	LDL	(−0.77, 4.9, 20.7)	(−1.36, 8.69, 37.37)	(0.93, 4.36, 20.82)			
Fruit and vegetable drink	TC	(4.86, 11.54, 28.72)	(8.43, 20.31, 52.57)	(0.19, 0.65, 2.88)	(0.67, 2.26, 10.38)		
	LDL	(−0.35, 2.32, 10.2)	(−0.63, 4.06, 18.37)				
	HCY	(−1.35, 1.47, 9.4)	(−2.4, 2.55, 17.02)				
	SBP	(−5.89, 2.33, 24.57)	(−10.29, 4.15, 44.62)	(−4.04, 1.58, 16.05)	(−14.24, 5.54, 57.98)	(−0.47, 0.16, 2.47)	(−1.63, 0.57, 8.86)
	BMI					(−0.48, 0.32, 2.6)	(−1.7, 1.11, 9.37)
Cherry juice	TC	(13.76, 21.01, 37.24)	(24.34, 36.8, 69.27)	(0.66, 1.33, 4.26)	(2.31, 4.69, 15.32)		
	SBP	(2.87, 11.88, 35.46)	(5.05, 20.71, 64.77)	(1.97, 7.94, 22.95)	(6.8, 27.7, 82.37)	(0.67, 3.01, 13.8)	(1.97, 7.94, 22.95)
Fruit juice	TC	(−7.46, 8.62, 46.75)	(−13.07, 15, 84.05)	(−0.3, 0.33, 3.37)	(−1.04, 1.15, 12.05)		
	LDL	(−1.71, 2.66, 15.16)	(−3.05, 4.65, 27.24)				

**Table 5 jcm-08-01914-t005:** Estimated effect in terms of the absolute number of event reductions and their associated direct costs for different supplementation regimes in subjects with hypertension, projected in 2025. The median and the 95% credibility interval are reported. Reported events are expressed in millions of events (95% C.I. lower bound; median; 95% C.I. upper bound), while direct costs are expressed in billions of dollars (95% C.I. lower bound; median; 95% C.I. upper bound).

		Coronary Heart Disease		Stroke		Heart Failure	
		Events	DIRECT COSTS	Events	Direct Costs	Events	Direct Costs
Encapsulated fruit and vegetable	TC	(−1.05, 2.46, 12.88)	(−3.09, 7.24, 38.8)	(−0.04, 0.08, 0.86)	(−0.07, 0.13, 1.56)		
	LDL	(−0.1, 0.16, 0.94)	(−0.29, 0.46, 2.85)				
	HCY	(40.6, 46.49, 64.47)	(118.17, 137.27, 202.7)				
	SBP	(4.7, 6.5, 12.36)	(13.67, 19.26, 38.28)	(3.51, 5.05, 10.63)	(6.17, 8.95, 19.49)	(1.65, 2.5, 6.2)	(5.7, 8.71, 22.2)
Artichoke leaf juice	TC	(15.46, 26.35, 54.12)	(45.46, 77.4, 166.81)	(0.42, 1.13, 4.76)	(0.73, 1.98, 8.47)		
	LDL	(−1.84, 0.99, 8.73)	(−5.44, 2.93, 26.21)				
	SBP	(13.96, 15.67, 20.78)	(40.53, 46.46, 66.33)	(9.99, 12.23, 18.73)	(17.39, 21.48, 34.61)	(4.61, 6.02, 11.58)	(16.02, 21.08, 42.03)
Jerusalem artichoke juice	TC	(−12.49, 4.61, 45.21)	(−36.99, 13.57, 136.06)	(−0.45, 0.11, 3.43)	(−0.8, 0.2, 6.17)		
	LDL	(−3.65, 1.72, 16.56)	(−10.74, 5.03, 50.32)				
Orange juice	TC	(19.13, 30.72, 60.45)	(56.28, 90.33, 186.29)	(0.52, 1.34, 5.59)	(0.93, 2.35, 10.05)		
	LDL	(6.32, 10.55, 23.15)	(18.69, 31.19, 70.81)				
Fruit and vegetable drink	TC	(3.91, 9.91, 27.14)	(11.44, 29.05, 81)	(0.08, 0.38, 2.06)	(0.14, 0.67, 3.75)		
	LDL	(−0.23, 1.78, 7.83)	(−0.68, 5.27, 23.83)				
	HCY	(−0.98, 1.08, 7.16)	(−2.88, 3.17, 21.93)				
	SBP	(−4.49, 1.65, 18.71)	(−13.4, 4.95, 56.42)	(−3.56, 1.31, 15.41)	(−6.24, 2.31, 27.83)	(−1.75, 0.62, 8.33)	(−6.04, 2.2, 29.83)
	BMI					(−1.03, 0.73, 6.64)	(−3.57, 2.53, 23.63)
Cherry juice	TC	(11.66, 18.36, 38.16)	(34.53, 54.33, 116.56)	(0.33, 0.79, 3.23)	(0.58, 1.4, 5.79)		
	SBP	(2.37, 8.86, 27.46)	(6.89, 25.98, 83.3)	(1.82, 6.75, 22.46)	(3.25, 11.86, 40.89)	(0.86, 3.24, 12.41)	(2.99, 11.36, 44.51)
Fruit juice	TC	(−4.83, 6.06, 33.34)	(−14.2, 18.05, 101.26)	(−0.17, 0.19, 2.47)	(−0.3, 0.33, 4.45)		
	LDL	(−1.41, 1.9, 11.52)	(−4.18, 5.62, 35.05)				

**Table 6 jcm-08-01914-t006:** Estimated effect in terms of the absolute number of event reductions and their associated direct costs for different supplementation regimes in subjects with metabolic syndrome, projected in 2025. The median and the 95% credibility interval are reported. Reported events are expressed in millions of events (95% C.I. lower bound; median; 95% C.I. upper bound), while direct costs are expressed in billions of dollars (95% C.I. lower bound; median; 95% C.I. upper bound).

	Coronary Heart Disease	Stroke	Heart Failure
Events	Direct Costs	Events	Direct Costs	Events	Direct Costs
Encapsulated fruit and vegetable	TC	(−0.98, 2.75, 13.15)	(−1.72, 4.81, 23.95)	(−0.03, 0.09, 0.86)	(−0.11, 0.31, 3.05)		
LDL	(−0.42, −0.15, 0.63)	(−0.74, −0.26, 1.12)				
HCY	(0.63, 0.74, 1.17)	(1.1, 1.32, 2.17)				
SBP	(4.53, 6.37, 12.56)	(7.94, 11.26, 22.98)	(2.94, 4.22, 8.81)	(10.29, 14.86, 31.86)	(0.96, 1.43, 3.29)	(3.32, 4.97, 12.12)
TNF-α	(1.94, 3.15, 6.75)	(3.39, 5.51, 12.29)	(1.34, 1.83, 2.57)	(4.68, 6.35, 9.83)		
Artichoke leaf juice	TC	(−11.85, 4.78, 43.02)	(−20.92, 8.47, 77)	(−0.44, 0.13, 3.34)	(−1.55, 0.45, 11.75)		
LDL	(−8.42, 3.86, 36.24)	(−14.77, 6.79, 65.79)				
Orange juice	TC	(18.34, 29.12, 57.4)	(32.14, 51.27, 104.71)	(0.58, 1.35, 5.38)	(2.03, 4.75, 19.43)		
LDL	(13.5, 22.47, 49.67)	(23.7, 39.79, 90.37)				
Fruit and vegetable drink	TC	(3.77, 9.51, 26.52)	(6.62, 16.69, 48.85)	(0.09, 0.37, 1.97)	(0.31, 1.31, 7.15)		
LDL	(−0.64, 3.95, 17.75)	(−1.1, 6.89, 31.88)				
HCY	(−0.93, 1.08, 7.02)	(−1.64, 1.9, 12.63)				
SBP	(−4.33, 1.68, 18.67)	(−7.7, 2.95, 33.56)	(−2.97, 1.13, 12.52)	(−10.42, 3.94, 44.67)	(−0.98, 0.38, 4.62)	(−3.44, 1.32, 16.85)
BMI					(−0.6, 0.43, 3.44)	(−2.09, 1.47, 12.09)
Cherry juice	TC	(11.2, 17.7, 37.3)	(19.6, 31.01, 68.63)	(0.35, 0.79, 3.03)	(1.23, 2.77, 10.86)		
SBP	(2.11, 8.46, 27.36)	(3.73, 14.85, 49.34)	(1.39, 5.56, 18.25)	(4.77, 19.37, 65.62)	(0.47, 1.84, 6.81)	(1.63, 6.44, 24.69)
Fruit juice	TC	(−4.71, 6.31, 34.51)	(−8.34, 11.02, 62.33)	(−0.17, 0.19, 2.4)	(−0.58, 0.66, 8.6)		
LDL	(−3.05, 4.18, 25.21)	(−5.3, 7.41, 46.22)

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
