# Peer review of "Fruit and Vegetable Concentrate Supplementation and Cardiovascular Health: A Systematic Review from a Public Health Perspective"

_jcm, 2019, doi:10.3390/jcm8111914_

Round 1

Reviewer 1 Report

Overall, I think the authors have done a tremendous job to improve the quality of this manuscript, in which case I greatly appreciate their efforts. I think this version is ready for publication. I would like to recommend for publication in Journal of Clinical Medicine. Thank you. 

Author Response

We would like to thank the reviewer for the comment.

Reviewer 2 Report

What is the main question addressed by the research?
The question is there is a health benefit to eating more fruits and vegetables and does supplementation have an effect.

Is it relevant and interesting?
Yes, the question is interesting because there are many papers on the importance of consuming fruits and vegetables but there is no comparison with supplementation. 

Is the document well written?
The document is very well documented and there is a lot of work to do.
However, this manuscript only exposes what can be read in the bibliography without comment. There is no argumentation on the results found.

Is the text clear and easy to read?
No, the text is not easy to read, there are large tables badly arranged, without keywords with only bibliographic references. These tables are incomprehensible. They need to detail them and explain the results obtained.

Are the conclusions consistent with the evidence and arguments presented? There is no real conclusion.

Do they answer the main question asked?
No, because there is no real synthesis work. In the end, we do not know whether consuming fruit and vegetables as a supplement is recommended for good health.

For Tables 2, 4 and 5: they are poorly arranged and not very comprehensible. You have to try to put keywords not only references.
You make large tables with lots of references, which is very good, but you don't detail them. You don't make any real comparisons. You do not provide conclusions or interpretation. You are missing a synthesis work. Tables alone are not enough.

Author Response

Is the document well written?
The document is very well documented and there is a lot of work to do.
However, this manuscript only exposes what can be read in the bibliography without comment.

The Discussion section has been improved, together with the Conclusions.

There is no argumentation on the results found.

The Discussion has been improved.

Is the text clear and easy to read?
No, the text is not easy to read, there are large tables badly arranged, without keywords with only bibliographic references. These tables are incomprehensible. They need to detail them and explain the results obtained.

The tables have been rearranged in order to improve the readability and further details have been added to the results section.

Are the conclusions consistent with the evidence and arguments presented? There is no real conclusion.

The conclusion section has been improved by providing a synthesis of the main findings of the study.

Do they answer the main question asked?
No, because there is no real synthesis work. In the end, we do not know whether consuming fruit and vegetables as a supplement is recommended for good health.

A summary of the main findings of the study has been provided at the beginning of the Discussion section. Furthermore, it has been clarified the take-home message of the study in the Conclusions section.

For Tables 2, 4 and 5: they are poorly arranged and not very comprehensible. You have to try to put keywords not only references.

Tables 2, 4, 5, and 6 have been rearranged as requested.

You make large tables with lots of references, which is very good, but you don't detail them. You don't make any real comparisons. You do not provide conclusions or interpretation. You are missing a synthesis work. Tables alone are not enough.

The results, the discussion, and the conclusions sections have been improved as requested.

Reviewer 3 Report

Introduction

Since the review focuses on cardiovascular outcomes, I suggest some background about relationships between CVD and FV intake, and including reference(s).

Line 67-68. Please provide some references to these studies of the effect of concentrates on human metabolism and health maintenance.

Line 74. Change ‘NCD’s to CVD, or at least specify that the review’s focus is CVD. The outcomes are not related to other NCDs, except possibly BMI.  Throughout the manuscript, the effect of the concentrates are discussed in the context of all NCDs, not CVD. This should be changed throughout; results only apply to CVD, not all NCDs.

Line 110.  Various definitions of metabolic syndrome specify the presences of at least 3 risk factors, and in some of them, insulin resistance must be one of them.  The metabolic syndrome group really only has 2 of the risk factors, since HDL and FBG were not among the outcomes, and hypertensives were analyzed separately. A comment about this should be included in the methods or discussion. 

Line 248. For the titles of Tables 4, 5, and 6, I suggest changing the word “gain” to “effect”.  If allowed by the journal, suggest putting the second sentence of the title as a footnote.

Figures 2, 3, and 4.   In lines 245-247, it says the results in terms of avoided CHD, stroke……   but the titles in the figures say number of cardiovascular events. Do the figures show actual numbers of events, or the reduction in the number of events? If it is reduction, revise titles to say “Reduction in number of…….”

The narrative of the results needs to be expanded.

There needs to be more discussion of the results.  For instance, is there literature about effect of fruit and vegetable intake on HCY? Is there some conflicting or supporting literature about effect of FV intake on some of the other outcomes?  More discussion about the results would strengthen conclusions.

Another limitation to consider is the difference in amounts, and unknown quantities of phytonutrients that might have been present.

Author Response

Comments and Suggestions for Authors

Introduction

Since the review focuses on cardiovascular outcomes, I suggest some background about relationships between CVD and FV intake, and including reference(s).

Done.

Line 67-68. Please provide some references to these studies of the effect of concentrates on human metabolism and health maintenance.

Done

Line 74. Change ‘NCD’s to CVD, or at least specify that the review’s focus is CVD. The outcomes are not related to other NCDs, except possibly BMI.  Throughout the manuscript, the effect of the concentrates are discussed in the context of all NCDs, not CVD. This should be changed throughout; results only apply to CVD, not all NCDs.

Done

Line 110.  Various definitions of metabolic syndrome specify the presences of at least 3 risk factors, and in some of them, insulin resistance must be one of them.  The metabolic syndrome group really only has 2 of the risk factors, since HDL and FBG were not among the outcomes, and hypertensives were analyzed separately. A comment about this should be included in the methods or discussion.

The choice of including or not the subjects in the group of those with metabolic syndrome was guided by the papers included in the systematic review, in order to guarantee homogeneous groups, even though not all clinical criteria have been included in the definition.

Line 248. For the titles of Tables 4, 5, and 6, I suggest changing the word “gain” to “effect”.  If allowed by the journal, suggest putting the second sentence of the title as a footnote.

Done. However, we thought that it is relevant to specify the nature of the study in the title (a systematic review from a public health perspective) since it highlights the value added by the study (using the data from the systematic review to simulate the public health impact of fruit and vegetable supplementation of the population).

Figures 2, 3, and 4.   In lines 245-247, it says the results in terms of avoided CHD, stroke……   but the titles in the figures say number of cardiovascular events. Do the figures show actual numbers of events, or the reduction in the number of events? If it is reduction, revise titles to say “Reduction in number of…….”

The figures refer to the number of events avoided. The labels have been changed accordingly. We would like to thank the reviewer for the comment.

The narrative of the results needs to be expanded.

Done

There needs to be more discussion of the results.  For instance, is there literature about effect of fruit and vegetable intake on HCY? Is there some conflicting or supporting literature about effect of FV intake on some of the other outcomes?  More discussion about the results would strengthen conclusions.

Done

Another limitation to consider is the difference in amounts, and unknown quantities of phytonutrients that might have been present.

Such issue was taken into account in the analyses and only single estimates for each class of products were provided, instead of pooled ones. The comment has been added to the study limitations section

Round 2

Reviewer 2 Report

Thank you for taking note of the remarks. Reading is easier and more enjoyable.

This manuscript is a resubmission of an earlier submission. The following is a list of the peer review reports and author responses from that submission.

Round 1

Reviewer 1 Report

The authors of this manuscript investigated the fruit and vegetable concentrate supplementation and cardiovascular health. This systematic review provides good insights of public health and disease prevention. I have some suggestions and questions to the authors.

The paper should also begin with some references targeting fresh fruits and vegetables. The reasoning behind this is that the readers need to know why there is a need to investigate concentrates instead of fresh fruits and vegetables (say, fresh fruits and vegetables are more affordable and convenient to purchase). Without this rationale, it is difficult to understand the need for fruits and vegetables concentrates.

I am including a few studies regarding this topic. Notice that these studies are related with sleep, but the rationale is that sleep is an important health behavior among animals and human beings. There is a relationship between human beings’ health condition, including NCDs, and sleep. You do not need to cite all of them, but I am only providing you some resources to strengthen the introduction section.

References:

Kruger AK, Reither EN, Peppard PE, Krueger PM, Hale L. Do sleep-deprived adolescents make less-healthy food choices? Br J Nutr. 2014;111(10):1898–904.

Duke CH, Williamson JA, Snook KR, Finch KC, Sullivan KL. Association between fruit and vegetable consumption and sleep quantity in pregnant women. Matern Child Health J. 2017;21(5):966–73.

Lee YH, Chang YC, Lee YT, Shelley M, Liu CT. Dietary patterns with fresh fruits and vegetables consumption and quality of sleep among older adults in mainland China. Sleep Biol Rhythms. 2018;16(3):293–305.

Tables 6, 7, and 8 are somewhat difficult to follow. Can authors either shorten these tables or make them concise?

On page 21, the paragraph between lines 265 and 267 is not necessary. Since this is already a long paper, the authors should try to be as concise as possible. The next paragraph already summarizes the purpose of the study, adding another paragraph on top of it makes this section wordy.

On page 22, the paragraph between lines 312 and 314 should be moved to the section of study limitations.

Also, I have another concern regarding the study. Since the groups of consumers who take fruits and vegetables concentrates or supplements regularly might have higher socioeconomic status, the study results could be affected by other factors such as their living environment, physical activity, and level of education. Although this is a systematic review, it should be careful that some studies might have endogeneity regarding this issue. Additionally, as the authors point out that some study samples are small with lower level of statistical power, such concerns are highly possible.

In general, I think this review has its significance. I hope these comments and questions can help the authors with their revisions. Thank you very much.

Reviewer 2 Report

«Fruit and vegetable concentrate supplementation and cardiovascular health: a systematic review from a public health perspective»

The objective of the present manuscript was to perform a systematic review -according to the PRISMA guidelines - with the aim of assessing the effects of fruit or vegetable concentrate supplementation on select physiological parameters that are considered relevant risk factors for the development of NCDs. The public health consequences of various scenarios of concentrate utilization are also examined in this paper.

General comments.

Although the topic is quite interesting and appealing, the manuscript lacks scientific justification.

The Material & methods section should provide information regarding: The probability function(s) that authors proposed to simulate, How the problem of autocorrelation was addressed, How the problem of the "curse of dimensionality" was addressed and, Whether the studies were weighted and if yes how.

Discussion: A revised version of the discussion section is suggested. Several parts, such as in lines 297 – 331, are often vague and long-winded.

Conclusions: If a brief conclusion, concise and into the point, should be included.

Special comments.

Line 58-60: Please rewrite in a more clear way.

Lines 60-61: “Supplementation … subgroups”. Although supplements could be an excellent alternative in case of low fruit/vegetable consumption the fact that they cannot replace a healthy and balanced diet should be highlighted.

Line 143: Please explain the abbreviation “HF”.

Line 157: What algorithm the authors used for MCMC Markov Chain Monte Carlo (MCMC)? Furthermore, why the authors used the MCMC method for simulation, instead of resampling or/and use other Bootstrap methods (that are more appropriate for Meta-Analysis)? The estimated 95% confidence regions for the various estimation have any logical and biological interpretation? Lines 265-267: These lines, should be omitted from the discussion section. This information is provided by the journal to facilitate the writing process and it is not presented in the final manuscript.

Lines 278-296: These lines, dealing with the effectiveness of supplementations by evaluating their action on specific biomarkers of metabolic and cardiac function, is rather wordy and should be rewritten in order to be more accurate and highly informative.

Lines 315 – 331: These lines are , in my opinion, quite irrelevant to the topic.

Lines 333-346. These lines should be rewritten in order to be more accurate and precise.

Lines 347-352: Please explain in more detail.

Lines 354-355: These lines should be omitted from the discussion section. This information is provided by the journal to facilitate the writing process and it is not presented in the final manuscript.

Tables

Table 2: This table should be moved to the supplementary materials.

Table 3: Please consider proving information regarding the composition of concentrates in each study. Do you believe that the fact that in some studies the concentrates were encapsulated could have affected the recorded results?

Table 4: This table should be moved to the supplementary materials.